# EMPOWERING GNNS FOR DOMAIN ADAPTATION VIA DENOISING TARGET GRAPH

## ABSTRACT

We explore the node classification task in the context of graph domain adaptation, which uses both source and target graph structures along with source labels to enhance the generalization capabilities of Graph Neural Networks (GNNs) on target graphs. Structure domain shifts frequently occur, especially when graph data are collected at different times or from varying areas, resulting in poor performance of GNNs on target graphs. Surprisingly, we find that simply incorporating an auxiliary loss function for denoising graph edges on target graphs can be extremely effective in enhancing GNN performance on target graphs. Based on this insight, we propose our framework, GRAPHDET, a framework that integrates this auxiliary edge task into GNN training for node classification under domain adaptation. Our theoretical analysis connects this auxiliary edge task to the graph generalization bound with $\mathcal{A}$-distance, demonstrating such auxiliary task can imposes a constraint which tightens the bound and thereby improves generalization. The experimental results demonstrate superior performance compared to the existing baselines in handling both time and regional domain graph shifts.

## 1 INTRODUCTION

Graph Neural Network (GNN) has shown success in learning on graph data on various web applications including social network Rossi et al. (2020); Fan et al. (2019), recommendation Wu et al. (2022); Lee et al. (2024), fraud detection Wang et al. (2019); Liu et al. (2021); He et al. (2022), etc. However, one major challenge is that real-world graph data continuously evolves over time, introducing changes in both graph structure and node/edge features between the training and testing graphs, which is a graph domain adaption problem. This may result in a substantial disparity between training and testing performance. However, collecting new labels and retraining models on the latest datasets is expensive or sometimes infeasible. For instance, it may take days or weeks for a customer to report a damaged or missing product, by which time the graph has already become outdated. Therefore, designing a GNN training method that can solve or alleviate this graph domain adaption challenges is essential for deploying GNNs in real-world applications.

However, the domain adaption problem on graph data is still under-explored. Unlike image data, graphs are non-grid data and inherently contain valuable structural information, which plays a crucial role in determining the final predictions. In particular, for node classification tasks, node labels are highly influenced by connectivity patterns within the graph structures (Huang et al., 2020). Therefore, domain adaptation methods should consider such structural information, making it challenging to directly apply existing computer vision techniques to graph domain adaption. Moreover, graph domain adaption has a wide application in real-world scenarios. For example, GNNs need to be trained on graphs collected from a specific time period with scarce and costly labels, while they are expected to have the ability to perform the classification tasks on future graphs where labels are unavailable. Similarly, labeled graphs may be collected from one region, while people expect trained GNNs to generalize well when applied to graphs gathered from other regions. However, variations in time and geographical domains inevitably introduce shifts in graphs, particularly in their structural information.

In this work, we propose our framework GraphDeT, incorporating the edge tasks on target graphs when training GNNs, improving the GNN generalization bound. We summarize our contributions here:

- We find that incorporating auxiliary edge tasks in our framework GRAPHDET is highly effective, demonstrating significant performance improvements over existing baselines. Specifically, the accuracy improvement on Arxiv is from 7.30% to 21.83%, and GRAPHDET achieves significant improvement on 8 out of 10 graph domain adaption tasks on MAG with improvement from 3.45% to 26.75%. This insight may provide valuable guidance for future method design in graph domain adaption problem.
- We extend the domain generalization bound by explicitly incorporating the structural information of graph data. By building the connection between this bound and our proposed auxiliary edge task, we demonstrate that the task can effectively constrain the $\mathcal{A}$-distance for graph domain generalization problem.

## 2 PRELIMINARY AND RELATED WORKS

### 2.1 PRELIMINARY

**Graph Data**, denoted as $\mathcal{G}$, is formally defined as $\mathcal{G} = (\mathcal{V}, \mathcal{E})$ with $n$ nodes. Specifically, $\mathcal{V} = \{v_1, \cdots, v_n\}$ represents a set of nodes containing $n$ elements with node features $\mathbf{X} \in \mathbb{R}^{n \times d}$, where $d$ is the dimension of input node features. The set of edges, denoted as $\mathcal{E}$, is formally defined as $\mathcal{E} \subseteq \mathcal{V} \times \mathcal{V}$. Typically, it can be represented by the adjacent matrix $\mathbf{A} \in \{0,1\}^{n \times n}$, which is a binary matrix. In this matrix, $\mathbf{A}_{ij} = 1$ if there exists an edge connecting node $v_i$ to $v_j$, and $\mathbf{A}_{ij} = 0$ otherwise. For the node classification task in this paper, each node has its label, denoted as $\mathbf{Y} \in \mathbb{N}^n$. For the graph domain adaption problem, source graphs also known as training graphs are separated from target graphs also known as test graphs. We use these settings in the following sections.

**Graph Neural Networks** (Kipf and Welling, 2017; Hamilton et al., 2017; Velickovic et al., 2018; Corso et al., 2020; Li et al., 2019; Gao and Ji, 2019) follow a message passing scheme (Scarselli et al., 2008; Gori et al., 2005). Given the hidden features from layer $l$, the hidden features in next layer $l + 1$ are defined as

$$\mathbf{h}^{(l+1)} = \text{UPDATE}\left(\text{AGGREGATE}(\mathbf{h}^{(l)}, \mathbf{A}); \mathbf{h}^{(l)}\right). \tag{1}$$

The aggregation module gathers the neighbors' information of each node using this adjacent matrix $\mathbf{A}$ in the aggregation module, and then the update module uses aggregated features to update the original node features $\mathbf{h}^{(l)}$ to get the new node features $\mathbf{h}^{(l+1)}$. The GNN model with $t$ message passing layers takes the graph $\mathcal{G}$ as the input of the model, denoted $f(\mathcal{G})$, and outputs the hidden feature $\mathbf{h}^{(t)}$. This hidden feature is then fed into a classifier $g$ to make the final prediction $g(\mathbf{h}^{(t)}) = g \circ f(\mathcal{G}) = \hat{Y}$, where $\hat{Y}$ is the predicted label. Besides, linear GNNs precompute the node features and then feed these precomputed features into the later Multi-Layer Perceptrons (MLPs) to learn the classifier. For example, Frasca et al. (2020); Chien et al. (2021); Wu et al. (2019); Lee et al. (2024) precompute the $k$-hop aggregated features using the variant adjacent matrix $\hat{\mathbf{A}}^k$, denoted as:

$$\Theta_k = \hat{\mathbf{A}}^k \mathbf{X}, \tag{2}$$

where $\Theta_k$ denotes the features aggregated from $k$-hops.

**Unsupervised Domain Adaption (UDA)** is a widely discussed problem in the computer vision domain (Zhang and Gao, 2022; Zhao et al., 2020; Wilson and Cook, 2020; Csurka, 2017). Especially for autonomous driving tasks (Schwonberg et al., 2023), where obtaining real-world labels across various conditions is challenging and costly, simulated environments provide a practical alternative by generating synthetic data that is easier to produce and label. Therefore, in these tasks, simulated synthetic images with labels will be treated as the source domain, and the model will be trained on these and then tested on the real-world data. Various works have explored the potential to apply adversarial methods (Xiao et al., 2024; Rangwani et al., 2022; Du et al., 2021; Wei et al., 2021), optimal transport (Nguyen et al., 2021; Fatras et al., 2021), pseudo-label based methods (Shin et al., 2020; Chen et al., 2020) to help improve the model trained on source data to have a better performance on test data. To analyze the domain adaption problem and provide theoretic guidance for the following methods, several works (Ben-David et al., 2006; 2010; Zhao et al., 2019) exploring the generalization upper bound for the domain adaption problem.

**Theorem 2.1 (Ben-David et al. (2006); Zhao et al. (2019)):** *Let $\mathcal{H}$ be a hypothesis space of VC-dimension $d$ and $\widehat{\mathcal{D}}_S$ (resp. $\widehat{\mathcal{D}}_T$ ) be the empirical distribution induced by a sample of size $n$ drawn from $\mathcal{D}_S$ (resp. $\mathcal{D}_T$ ). Then w.p. at least $1 - \delta, \forall h \in \mathcal{H}$,*

$$\epsilon_T(g) \le \widehat{\epsilon}_S(g) + \lambda^* + O\left(\sqrt{\frac{d\log n + \log(1/\delta)}{n}}\right)$$

$$+ \frac{1}{2}d_{\mathcal{H}\Delta\mathcal{H}}\left(\widehat{\mathcal{D}}_S, \widehat{\mathcal{D}}_T\right), \tag{3}$$

*where $\epsilon_T$ denotes the expected error on target domain, and $\hat{\epsilon}_S$ denotes the empirical error on source domain. The classifier $g$ belongs to hypothesis space $H$.*

In this bound equation, the $\mathcal{A}$-distance is denoted as

$$d_{\mathcal{H}\Delta\mathcal{H}}\left(\widehat{\mathcal{D}}_S, \widehat{\mathcal{D}}_T\right) = \left|\Pr_{\mathcal{D}_S}\left[\mathcal{Z}_{g_1}\Delta\mathcal{Z}_{g_2}\right] - \Pr_{\mathcal{D}_T}\left[\mathcal{Z}_{g_1}\Delta\mathcal{Z}_{g_2}\right]\right|,$$

where $\mathcal{Z}_g = \{\mathbf{z} \in \mathcal{Z} : g(\mathbf{z}) = 1\}$, $\mathcal{Z}$ is the input embedding space, $g_1$ is the best classifier learned from source domain, and $g_2$ learns from both domains, denoted as $g_2 = \arg\max_{g\in H}(\epsilon_S(g) + \epsilon_T(g))$. The $\lambda^* = \lambda_S + \lambda_T$ which represent the summation of errors of $g_2$ on source and target data. Moreover,

$$\mathcal{Z}_{g_1}\Delta\mathcal{Z}_{g_2} = \left\{\begin{array}{ll} 1 & \text{if} \quad g_1(Z) \neq g_2(Z), \\ 0 & \text{if} \quad g_1(Z) = g_2(Z) \end{array}\right.$$

This above error bound has a more simpler version (Ben-David et al., 2006),

$$\epsilon_T(h) \le \lambda_T + \Pr_{\mathcal{D}_T}\left[\mathcal{Z}_{g_1}\Delta\mathcal{Z}_{g_2}\right] \tag{4}$$

Intuitively, this bound is related to the different prediction results between the classifiers $g_1$ and $g_2$.

In this work, we focus on the node classification problem, where we have a source graph $\mathcal{G}_S = (\mathcal{V}_S, \mathcal{E}_S)$ and a target graph $\mathcal{G}_T = (\mathcal{V}_T, \mathcal{E}_T)$, with unpaired nodes between these two graphs. The source graph $\mathcal{G}_S$ comes with node labels $Y_S$, while the target graph $\mathcal{G}_T$ is unlabeled during training. Compared to prior UDA approaches, in this work, we emphasize that the structural information of both source and target graphs are important for tightening the generalization bound for graph UDA problems.

## 2.2 RELATED WORKS

**Graph domain adaption** (Shi et al., 2024; Dai et al., 2022; Shen et al., 2023) is challenging when training GNNs on a source graph and testing on target graphs due to distribution shifts between them. Various works (Mao et al., 2024; Pang et al., 2023; Cai et al., 2024; Zhang et al., 2019; Wu et al., 2020; Zhu et al., 2021; 2022; Fang et al.) have been explored to tackle this problem. StruRW and Pairwise Alignment (PA) (Liu et al., 2023) study conditional structure shifts (CSS) and label shifts, using pseudo labels to calculate edges weights alleviating CSS during training for graph domain adaption.

SpecReg (You et al., 2023) analyzes the domain adaptation generalization bound using optimal transport and introduces a regularization term on graph spectrum to constrain the bound's constant coefficients. A2GNN (Liu et al., 2024a) further tightens the bound of the constant coefficients in SpecReg shown in Lemma3 of A2GNN. Compared to SpecReg and A2GNN, our framework leverages an auxiliary edge-prediction task to explicitly constrain the $\mathcal{A}$-distance. As stated in Section 4.1 of SpecReg, "Other data-relevant properties (e.g. $W1(PS(G), PT(G))$) are left to future works". This part is related to A-distance, which naturally captures the data-relevant property. Our proof builds on this insight and represents a step forward from prior work in graph domain generalization by directly constraining the $\mathcal{A}$-distance. Although the proposed auxiliary edge tasks are simple and straightforward, they provide an effective mechanism for constraining the generalization bound while incorporating structural information, and our work offers the theoretical justification for why this approach is effective.

Figure 1: The pipeline of our proposed GRAPHDET. For the graph domain adaption task, both source graphs and target graphs are used. In the upper branch, the source graph is fed into the GNN model get node embeddings which are applied to MLPs for the class prediction tasks with the labels from source graph. In the bottom branch, the target graphs are attached with random noisy edges and then fed into the shared GNN architectures. The obtained embeddings are then fed into MLPs to perform classification on these edges.

## 3 METHOD - GRAPHDET

In this section, we firstly present several data analyses using the Arxiv dataset and illustrate the intuition of leveraging link tasks on target graphs for enhancing graph domain adaptation generalization performance. Next, based on these observations, we propose GRAPHDET.

### 3.1 PROPOSED GRAPHDET AND ITS VARIATIONS

In this subsection, we introduce GRAPHDET, which is simple and straightforward. It integrates auxiliary edge tasks trained alongside the classification loss on source graphs. After the details about this framework, we then provide a theoretical justification to provide insights for its effectiveness on graph domain adaptation tasks.

First, we show one of the potential edge task aiming to denoise the target graphs within our framework. Specifically, we first add random edges on the target graph, denoted as:

$$\tilde{\mathbf{A}}_T = \mathbf{A}_T + \mathbf{A}_T^{'}, \tag{5}$$

where $\mathbf{A}_T^{'}$ is a sparse matrix randomly sampled from the possible node pairs, treated as the negative edges. This noisy graph is fed into GNNs to get the node embeddings which are served to perform the link classification task to detect which edges are noisy edges:

$$\tilde{\mathbf{h}}^T = f\left(\tilde{\mathcal{G}}_T(\tilde{\mathbf{A}}_T, \mathbf{X}_T)\right)$$
$$\tilde{s}_{uv} = \sigma\left(\phi(\tilde{\mathbf{h}}_u^T \circ \tilde{\mathbf{h}}_v^T)\right), \tag{6}$$

where $u, v$ are two nodes, $\circ$ denotes the Hadamard product and $\phi$ denotes the MLPs for edge prediction. For the edge denoising loss function, it is defined as:

$$\ell_{\text{DeT}} = \mathbb{E}_{(u,v)\sim\mathbf{A}_T}\log \tilde{s}_{uv} + \mathbb{E}_{(u,v)\sim\mathbf{A}_T^{'}}\log(1-\tilde{s}_{uv}), \tag{7}$$

which performs edge classification task on the input noisy target graphs to figure out the real edges within the original target graphs. The training procedure also contains the classification loss function on source graphs:

$$\ell_{\text{cls}} = D_{\text{KL}}(\mathbf{Y}_S\|\text{softmax}(g \circ f(\mathcal{G}_S(\mathbf{A}_S, \mathbf{X}_S)))). \tag{8}$$

During training, the GNNs are trained with both loss functions and the overall pipeline is demonstrated in Figure 1. Note that there are various edge tasks. For example, we can take use of the

GAE (Kipf and Welling, 2016), taking all edges as input and reconstructing these edges. We can also use link prediction tasks to split part of edges as input of target graphs, and predict the remaining edges (Zhang and Chen, 2018). We compare these multiple potential edge tasks in Section 4.2.

Selecting the edge denoising task for target graphs offers several intuitive advantages. Firstly, unlike traditional link prediction tasks that require withholding only a portion of edges during training, edge denoising retains all edges as input. This approach ensures that the GNNs are exposed to the complete graph structure, facilitating a more comprehensive learning process. Secondly, compared to GAE, our method, GRAPHDET, deliberately introduces fake edges into the target graphs. This intentional addition prevents the model from merely memorizing existing connections. Instead, it compels the model to learn to distinguish between original and fake graph patterns, thereby enhancing its ability to generalize and recognize meaningful structures within the data.

---

**Algorithm 1** GRAPHDET: graph domain adaption with auxiliary edge tasks on target graphs, e.g. denoising target graphs

---

**Require:** Input source graph $\mathcal{G}_S(\mathbf{A}_S, \mathbf{X}_S)$ with labels $\mathbf{Y}_S$ and target graph $\mathcal{G}_T(\mathbf{A}_T, \mathbf{X}_T)$.
**Ensure:** w
  1: Randomly initialize the model parameters $\mathbf{w}$
  2: **for** epoch $= 0$ to $N$ **do**
  3:     Add random edges on target graphs according to Eqn. equation 5
  4:     Calculate the edge score on all true and fake node pairs according to Eqn. equation 6
  5:     Get $\hat{Y}_S = g \circ f(\mathcal{S}_S)$ on source graph
  6:     Calculate the denoising target graph loss according to Eqn. equation 7
  7:     Calculate the node classification loss according to Eqn. equation 8
  8:     Update the model parameters $\mathbf{w}$ with optimizer
  9: **end for**

---

## 3.2 THEORETICAL JUSTIFICATION

In this section, we present the theoretical justification for how auxiliary edge tasks on target graphs contribute to solving the graph domain adaptation problem.

**Auxiliary Edge Tasks and Generalization Bound.** The auxiliary edge-prediction task aims to make embeddings of connected nodes in the target graph similar in the embedding space. Since the downstream classifier is Lipschitz continuous, this similarity in embeddings translates into small differences in the classifier outputs for connected nodes. Consequently, the disagreement between the classifier trained on the source graph ($g_1$) and the hypothetical classifier trained on the target graph ($g_2$) is reduced. A smaller disagreement score directly tightens the $\mathcal{A}$-distance term in the domain adaptation generalization bound, leading to improved generalization. We provide the proof in Appendix A for the proposition 3.1.

**Proposition 3.1 (Generalization Bound for GRAPHDET on Graph Domain Adaption Task):**
*For the graph domain adaption problem, let $\mathcal{R} : \mathcal{G}(\mathbf{A}, \mathbf{X}) \to \mathcal{Z}$ be a fixed representation function mapping from target graphs to a latent space. Let $g_1$ and $g_2$ to be two classifiers defined on $\mathcal{Z}$, and $g_1 = \arg\min_g \epsilon_S(g)$, $g_2 = \arg\min_g(\epsilon_S(g) + \epsilon_T(g))$. Meanwhile, $g_1$ and $g_2$ are $L_1$ and $L_2$-Lipschitz continuous, respectively. Let $C_{u^*}$ denote a connected component in target graph with node $u^*$ have the lowest $d(g_1(u^*), g_2(u^*))$, $N_{C_{u^*}}$ denotes the number of nodes within this connected component and $N$ denotes the number of nodes in target graph, and $K_{avg}(C_{u^*})$ be the average hop distance between the nodes within $C_{u^*}$ and $u^*$. Let $\xi_1$ be the constrains of l2-distance on node embeddings along any edge and $\xi_2$ be the constrain on the disagreement scores. Under these conditions, the $\mathcal{A}$-distance between $g_1$ and $g_2$ on $\mathcal{D}_T$ is bounded with probability $1 - \delta$ by*

$$\mathrm{Pr}_{D_T}\left(Z_{g_1} \Delta Z_{g_2}\right) \leq \frac{1}{\xi_2} \sum_{C_{u^*}(\mathcal{G}_T)} \frac{N_{C_{u^*}(\mathcal{G}_T)}}{N}$$

$$(d\left(g_1\left(x_{u^*}\right), g_2\left(x_{u^*}\right)\right) + K_{avg}(C_{u^*})(L_1 + L_2)\xi_1). \tag{9}$$

With the proposition 3.1, we can get that when we constrain the node embeddings along the edges to be similar, which is one of the goals of these edge tasks. The upper bound tends to be smaller

Table 1: Statistics and properties of the datasets.

| Dataset | # of nodes | # of edges | Avg. degree | # of features | # of classes | Train/Val/Test | # of dummy nodes | # of edges connected to dummy nodes |
|---|---|---|---|---|---|---|---|---|
| Arxiv (1950, 2007) | 4,980 | 11,698 | 2.35 | 128 | 40 | 2,988/-/- | - | - |
| Arxiv (1950, 2009) | 9,410 | 26,358 | 2.80 | 128 | 40 | 5,646/-/- | - | - |
| Arxiv (1950, 2011) | 17,401 | 60,972 | 3.50 | 128 | 40 | 10,440/-/- | - | - |
| Arxiv (2014, 2016) | 69,499 | 464,838 | 6.69 | 128 | 40 | -/5,674/22,700 | - | - |
| Arxiv (2016, 2018) | 120,740 | 1,230,830 | 10.19 | 128 | 40 | -/10,248/40,993 | - | - |
| MAG US | 132,558 | 1,394,900 | 10.52 | 128 | 20 | 79,534/14,123/56,586 | 61,849/(0.47) | 716,612/(0.51) |
| MAG CN | 101,952 | 571,122 | 5.60 | 128 | 20 | 61,171/7,703/30,318 | 63,931/(0.63) | 420,096/(0.74) |
| MAG DE | 43,032 | 253,326 | 5.88 | 128 | 20 | 25,819/4,326/17,288 | 21,418/(0.50) | 146,738/(0.58) |
| MAG JP | 37,498 | 181,888 | 4.85 | 128 | 20 | 22,498/2,969/11,988 | 22,541/(0.60) | 130,026/(0.71) |
| MAG RU | 32,833 | 135,988 | 4.14 | 128 | 20 | 19,699/992/3,882 | 27,959/(0.85) | 125,278/(0.92) |
| MAG FR | 29,262 | 156,444 | 5.35 | 128 | 20 | 17,557/2,765/11,152 | 15,345/(0.52) | 87,314/(0.56) |
| ogbn-products | 2,449,029 | 61,859,140 | 25.26 | 100 | 47 | 0.100/0.020/0.880 | - | - |

as $\xi_1$ becomes smaller if $d\left(g_1\left(x_{u^*}\right), g_2\left(x_{u^*}\right)\right)$ is fixed. That is the motivation why such edge tasks can help improve the generalization ability of graph domain adaption. However, we will also realize that if all the node embeddings on target graphs are the same, the node embeddings do not contain any information, and this bound is meaningless as $d\left(g_1\left(x_{u^*}\right), g_2\left(x_{u^*}\right)\right)$ will be the same for all nodes. Therefore, at the same time, we need to consider the other goal of these edge tasks. Usually, they will ensure the embeddings of connected nodes have high similarity while the nodes that are far away have low similarity. These edge tasks are trying to extract informative node embeddings considering the graph structures, and force the node embeddings to be different. As widely discussed in Xie et al. (2022), we can treat these edge tasks as a loss to extract information to avoid that the node embeddings collapse to the same embeddings. When the node embeddings can capture informative structural information, with the inherent relationship between source and target graphs, we can expect $d\left(g_1\left(x_{u^*}\right), g_2\left(x_{u^*}\right)\right)$ yield reasonable values for large connected components within the target graphs.

In conclusion, the edge tasks should consider both perspectives. Trying to extract informative structural information to avoid the collapse of node embeddings, and make the node embeddings along the edges to be similar to tighten the generalization upper bound.

## 4 EXPERIMENTS

**Software and Hardware.** The code implementatioin is based on PyTorch (Paszke et al., 2019), Pytorch_geometric (Fey and Lenssen, 2019) and DGL (Wang, 2019). Moreover, we conduct our experiments on NVIDIA A10G with 24GB memory, and AMD EPYC 7R32 CPU with 3271 MHz.

### 4.1 GRAPHDET

**Dataset.** We evaluate our proposed algorithm, GRAPHDET, through extensive graph domain adaptation node classification tasks. These include the time-domain adaptation dataset Arxiv (Hu et al., 2021; Liu et al., 2024c) and the regional-domain adaptation dataset MAG (Wang et al., 2020; Hu et al., 2021; Liu et al., 2024c). For the Arxiv dataset, there are three experimental settings. The source graphs consist of papers published from 1950 to 2007, 1950 to 2009, and 1950 to 2011, while the testing is conducted separately on target graphs containing papers published from 2014 to 2016 and from 2016 to 2018. For the MAG dataset, graphs are collected from six different countries: the US, China (CN), Germany (DE), Japan (JP), Russia (RU), and France (FR). The experiment will take one graph as the source graph to perform training and then test on target graphs collected in the other countries. Additionally, we use the ogbn-products dataset (Hu et al., 2021) for further investigation into semi-supervised learning tasks. In this setting, the training and test sets are split based on different sales rankings, introducing a distribution shift. Detailed statistics for these datasets are presented in Table 1.

**Baselines.** We compare our methods with the following baselines: ERM, which denotes the training of GNNs by minimizing the empirical risk and serves as a standard baseline to illustrate the outcome when no specific techniques are applied to address the domain adaptation problem; DANN (Ganin et al., 2016); IWDAN (Tachet des Combes et al., 2020); UDAGCN (Wu et al., 2020); SPECREG (You et al., 2023); StruRW (Liu et al., 2023); and Pairwise Alignment (PA) (Liu et al., 2024c).

Table 2: Comparison between our proposed GRAPHDET and other baselines on Arxiv dataset. The reported results are based on three random runs with accuracy scores as metric. The top performance is highlighted with **bold font**. Underline indicates that the corresponding methods have the second best performance.

| **DOMAINS** | 1950 − 2007 | | 1950 − 2009 | | 1950 − 2011 | |
| --- | --- | --- | --- | --- | --- | --- |
| | 2014 − 2016 | 2016 − 2018 | 2014 − 2016 | 2016 − 2018 | 2014 − 2016 | 2016 − 2018 |
| ERM | $37.91 \pm 0.31$ | $35.22 \pm 0.71$ | $43.50 \pm 0.35$ | $40.19 \pm 3.62$ | $51.76 \pm 0.93$ | $52.56 \pm 1.06$ |
| DANN | $37.31 \pm 1.54$ | $36.84 \pm 1.40$ | $43.57 \pm 0.47$ | $42.04 \pm 2.70$ | $53.02 \pm 0.67$ | $52.69 \pm 1.26$ |
| IWDAN | $36.16 \pm 2.91$ | $25.48 \pm 9.77$ | $41.26 \pm 2.08$ | $35.91 \pm 4.28$ | $46.73 \pm 0.62$ | $42.70 \pm 3.21$ |
| UDAGCN | $38.10 \pm 1.62$ | OOM | $42.85 \pm 2.09$ | OOM | $53.13 \pm 0.31$ | OOM |
| SPECREG | $37.09 \pm 0.62$ | $33.46 \pm 0.83$ | $43.14 \pm 2.16$ | $43.06 \pm 1.09$ | $52.63 \pm 1.29$ | $52.46 \pm 0.83$ |
| StruRW | $38.56 \pm 0.77$ | $37.17 \pm 2.75$ | $43.55 \pm 2.37$ | $43.55 \pm 2.37$ | $53.19 \pm 0.45$ | $53.64 \pm 0.65$ |
| PA-BOTH | $39.98 \pm 0.77$ | $40.23 \pm 0.30$ | $44.60 \pm 0.42$ | $44.43 \pm 0.34$ | $53.56 \pm 0.98$ | $51.60 \pm 0.24$ |
| GRAPHDET | $\mathbf{50.51 \pm 2.35}$ | $\mathbf{51.47 \pm 1.16}$ | $\mathbf{52.19 \pm 2.08}$ | $\mathbf{53.01 \pm 2.26}$ | $\mathbf{57.78 \pm 0.57}$ | $\mathbf{58.16 \pm 1.02}$ |
| Improvement | 20.84% | 21.83% | 14.54% | 16.18% | 7.30% | 7.78% |

**Setup.** We first employ the GraphSAGE (Hamilton et al., 2017) model within our framework to conduct experiments on both the Arxiv and MAG datasets. All other baselines follow previous works (Liu et al., 2024c), with the same GNN architecture. Specifically, we use a three-layer GNN with a hidden feature size of 300, followed by a two-layer MLPs as classifier with hidden features of 200 and a two-layer MLPs as link predictor with hidden features of 80. Additionally, the Hadamard product is applied to fuse the node features for each node pair in the link predictor. There is no normalization or residual connection within the GNNs following the previous works. For the target graph denoising procedure, we set the number of sampled negative edges to be the same as the edges in the original target graphs, ensuring a balanced distribution of real and fake node pairs. For the training hyper-parameters, we select the learning rate from $\{0.1, 0.03, 0.01, 0.005, 0.001, 0.0005\}$ and dropout from $\{0, 0.1, 0.2, 0.3, 0.5\}$. Meanwhile, for the weight decay, we explore from the range $\{1e - 3, 5e - 4, 1e - 4\}$. Moreover, for the MAG datasets, we also apply the Pairwise Alignment (PA) within our framework. Thus, we also explore the hyper-parameters of PA following the previous parameter selection (Liu et al., 2024c). Specifically, we select $\lambda_\beta$ from $\{0.005, 0.01\}$ related to alleviating the label shifting, $\delta$ from $\{1e - 4, 1e - 3\}$ related to the degree of structure shift, $\lambda_w$ from $\{1, 5, 10\}$ related to the conditional structure shift.

Our experiments are conducted over three runs with different random seeds. The final results are reported as the average accuracy on the target graphs along with the standard deviation. Additionally, in the MAG dataset, there is a dummy class, indicating that all nodes assigned to this class belong to other unknown classes. Therefore, following previous settings, the accuracy scores on test graphs are computed excluding nodes belonging to this dummy class, while during training, these nodes are treated the same as all other nodes.

**Results.** First, we evaluate the performance of our proposed methods on the time-domain adaptation dataset Arxiv. The corresponding accuracy on the target dataset is presented in Table 2. We observe a significant performance improvement in our methods compared to other baselines. Specifically, when the time gap is large, such as training on Arxiv (1950–2007) and testing on the target graph from Arxiv (2016–2018), our approach achieves a substantial improvement of **21.83%** over the second-best baseline, PA-BOTH, with a **11.24** accuracy score increase. As the time difference decreases, meaning that the source graph includes papers that are closer to the target graphs, we observe that the performance of both the baselines and our method improves as the distribution shift reduces. Although the improvement ratio becomes smaller, our method still achieves a notable gain of **7.30%** on the source graph Arxiv (1950–2011) and target graph Arxiv (2014–2016) compared to the second-best baseline, PA-BOTH, with an accuracy score improvement of **4.22**.

Then, we evaluate the performance of our proposed methods on the regional domain adaptation dataset MAG. The corresponding accuracy on the target dataset is presented in Table 3. When analyzing the graph data statistics, we find that dummy nodes have a significant impact on the MAG dataset. As shown in Table 1, a large proportion of nodes within the graphs are dummy nodes, and the edges connected to these dummy nodes dominate the overall graph connectivity. For instance, in the RU dataset, dummy nodes account for 85% of all nodes, while 92% of the edges are connected to dummy nodes. Notably, the proportion of dummy nodes and the ratio of edges connected to them

Table 3: Comparison of performance on MAG datasets with accuracy scores. The bold font and underline indicate the best model and second best performance respectively. We compare the methods with CSS and without CSS, individually.

| Domains | US → CN | US → DE | US → JP | US → RU | US → FR |
|---|---|---|---|---|---|
| ERM | $26.92 \pm 1.08$ | $26.37 \pm 1.16$ | $37.63 \pm 0.36$ | $21.71 \pm 0.38$ | $20.11 \pm 0.34$ |
| DANN | $24.20 \pm 1.19$ | $26.29 \pm 1.44$ | $\underline{37.92 \pm 0.25}$ | $21.76 \pm 1.58$ | $20.71 \pm 0.29$ |
| IWDAN | $23.39 \pm 0.93$ | $25.97 \pm 0.41$ | $34.98 \pm 0.68$ | $\underline{22.80 \pm 3.03}$ | $\underline{21.75 \pm 0.81}$ |
| UDAGCN | OOM | OOM | OOM | OOM | OOM |
| SPECREG | $23.74 \pm 1.32$ | $\underline{26.68 \pm 1.44}$ | $37.68 \pm 0.25$ | $21.47 \pm 0.84$ | $20.91 \pm 0.53$ |
| GRAPHDET | $\mathbf{32.42 \pm 0.51}$ | $\mathbf{34.75 \pm 1.62}$ | $\mathbf{45.01 \pm 0.43}$ | $\mathbf{30.60 \pm 1.58}$ | $\mathbf{28.04 \pm 1.03}$ |
| Improvement | 16.96% | 23.22% | 15.75% | 25.50% | 22.43% |
| STRURW | $31.58 \pm 3.10$ | $30.03 \pm 2.23$ | $37.20 \pm 0.27$ | $28.97 \pm 2.98$ | $22.73 \pm 1.73$ |
| PA-BOTH | $\mathbf{40.06 \pm 0.99}$ | $\underline{38.85 \pm 4.71}$ | $47.43 \pm 1.82$ | $37.07 \pm 5.28$ | $\underline{25.21 \pm 3.79}$ |
| GRAPHDET + PA-BOTH | $\underline{39.92 \pm 0.60}$ | $\mathbf{40.24 \pm 0.48}$ | $\mathbf{51.55 \pm 1.47}$ | $\mathbf{40.24 \pm 0.48}$ | $\mathbf{32.29 \pm 0.25}$ |
| Improvement | - 0.35% | 3.45% | 7.99% | 7.88% | 21.93% |
| **Domains** | **CN → US** | **CN → DE** | **CN → JP** | **CN → RU** | **CN → FR** |
| ERM | $31.47 \pm 1.25$ | $13.29 \pm 0.36$ | $\underline{22.15 \pm 0.89}$ | $10.92 \pm 0.82$ | $20.11 \pm 0.34$ |
| DANN | $30.23 \pm 0.99$ | $13.46 \pm 0.40$ | $21.48 \pm 1.26$ | $\underline{11.94 \pm 1.90}$ | $20.71 \pm 0.29$ |
| IWDAN | $\underline{31.72 \pm 1.24}$ | $13.39 \pm 1.06$ | $19.86 \pm 1.21$ | $10.93 \pm 1.33$ | $\mathbf{21.75 \pm 0.81}$ |
| UDAGCN | OOM | OOM | OOM | OOM | OOM |
| SPECREG | $26.52 \pm 1.75$ | $\underline{13.76 \pm 0.65}$ | $20.50 \pm 0.08$ | $10.50 \pm 0.53$ | $20.91 \pm 0.53$ |
| GRAPHDET | $\mathbf{42.08 \pm 0.57}$ | $\mathbf{27.18 \pm 1.21}$ | $\mathbf{34.45 \pm 2.00}$ | $\mathbf{22.35 \pm 1.10}$ | $\underline{21.04 \pm 0.17}$ |
| Improvement | 24.61% | 49.37% | 35.70% | 46.58% | -3.25% |
| STRURW | $37.08 \pm 1.09$ | $19.93 \pm 1.82$ | $29.76 \pm 2.56$ | $17.94 \pm 9.82$ | $22.73 \pm 1.73$ |
| PA-BOTH | $\underline{45.16 \pm 0.50}$ | $26.19 \pm 1.01$ | $38.26 \pm 2.27$ | $33.34 \pm 1.94$ | $25.21 \pm 3.79$ |
| GRAPHDET + PA-BOTH | $\mathbf{45.42 \pm 2.11}$ | $\mathbf{31.55 \pm 2.50}$ | $\mathbf{43.28 \pm 3.11}$ | $\mathbf{36.68 \pm 1.59}$ | $\mathbf{34.42 \pm 2.51}$ |
| Improvement | 0.57% | 16.9% | 11.60% | 9.11% | 26.75% |

Table 4: Comparison of various edge tasks on Arxiv.

| SOURCE | TARGET | ERM | GAE Loss | Link Prediction Loss | DeT Loss |
|---|---|---|---|---|---|
| $1950 - 2007$ | $2014 - 2016$ | $37.91 \pm 0.31$ | $47.91 \pm 0.96$ | $48.82 \pm 1.47$ | $50.51 \pm 2.35$ |
| | $2016 - 2018$ | $35.22 \pm 0.71$ | $49.90 \pm 2.58$ | $50.17 \pm 0.76$ | $51.47 \pm 1.16$ |
| $1950 - 2009$ | $2014 - 2016$ | $43.50 \pm 0.35$ | $50.71 \pm 3.24$ | $49.09 \pm 4.27$ | $52.19 \pm 2.08$ |
| | $2016 - 2018$ | $40.19 \pm 3.62$ | $47.19 \pm 6.64$ | $45.94 \pm 7.67$ | $53.01 \pm 2.26$ |
| $1950 - 2011$ | $2014 - 2016$ | $51.76 \pm 0.93$ | $57.29 \pm 0.53$ | $57.13 \pm 0.17$ | $57.78 \pm 0.57$ |
| | $2016 - 2018$ | $52.56 \pm 1.06$ | $57.80 \pm 0.75$ | $58.08 \pm 0.08$ | $58.16 \pm 1.02$ |

vary significantly across different graphs, ranging from 47% to 85% for dummy nodes and from 51% to 92% for edges. This unavoidable difference on the influence of dummy nodes leads to a significant conditional structure shift (CSS), as discussed in (Liu et al., 2023; 2024c). Therefore, we compare our methods with baselines that do not incorporate specific techniques to address this issue. Additionally, we integrate PA-BOTH within our framework and evaluate its final performance against StruRW and PA-BOTH. We can find a clear performance improvement compared to all the other baselines on most regions without alleviating CSS techniques. Specifically, the improvement of domain adaption from US to other five regions from **15.75% to 25.5%**. For domain adaptation from CN to other areas, the improvement is significant, increasing from **24.61% to 46.58%**, except in the case of adaptation from CN to FR. Notably, while our method does not enhance performance for CN-to-FR adaptation, applying the CSS reduction technique PA-BOTH with our method boosts the improvement to 26.75%, indicating that such adaption relies on reduction CSS technique a lot and our approach is particularly effective when addressing CSS-related challenges. Furthermore, incorporating PA-BOTH enhances the overall performance of our method. While the adaptation between the US and CN with PA-BOTH shows similar results with or without our approach, our method significantly improves performance compared to the previous PA-BOTH without auxiliary edge tasks, increasing from **3.45% to 26.75%** on all the other adaption settings. Above all, we believe the improvements on both time and regional domain adaption settings are impressive.

Table 5: Performance on the ogbn-products dataset.

| Methods | ERM | GRAPHDET |
|---|---|---|
| GraphSAGE | $78.28 \pm 0.16$ | $80.02 \pm 0.23$ |
| SIGN | $80.52 \pm 0.16$ | $81.14 \pm 0.13$ |

## 4.2 ABLATION STUDIES ON VARIOUS EDGE TASKS

In this section, we explore three different edge tasks within our framework. As discussed in section 3.1, there are multiple ways to implement edge tasks that achieve similar goals discussed in section 3.2. Specifically, we conduct our experiments following the previous settings on Arxiv datasets, and the final results are reported in Table 4. We can observe that all of these edge tasks can improve the model performance on these datasets. When the time difference is close, these methods do not have a clear difference on the performance, such as source graph Arxiv (1950-2011) and target graph Arxiv (2014-1016). When the time difference is a little larger, such as source graph Arxiv (1950, 2007) and target graph Arxiv (2016-2018) and the case source graph Arxiv (1950-2009) and target graph Arxiv (2016-2018), DeT Loss is constantly better than the other two.

## 4.3 STUDY ON SEMI-SUPERVISED LARGE-SCALE DATASET WITH DISTRIBUTION SHIFTING

The domain adaptation settings are similar to those of the semi-supervised learning task with distribution shifting. Both tasks use training (source) and testing (target) graphs during training, as well as the labels from the training graphs. The key difference lies in the connectivity between training and testing nodes. In semi-supervised learning settings, nodes in the training and testing sets are interconnected by edges, whereas in graph domain adaptation tasks, such edges typically do not exist. Given their similarities, we choose to evaluate our method on the large-scale, real-world ogbn-products dataset, which consists of millions of nodes and edges. The dataset is split based on sales ranking, with high sales ranking nodes included in the training set. This leads to a distribution shift, making it particularly interesting to explore whether our framework can enhance generalization on test graphs under these conditions.

**Setup.** We employ GraphSAGE Hamilton et al. (2017) and SIGN Frasca et al. (2020) model within our framework to conduct experiments on the ogbn-products dataset. Specifically, for GraphSAGE, we use a three-layer SAGE with a hidden feature size of 256 followed by a two-layer MLPs as classifier and another two-layer MLPs as link predictor with hidden features of 256. [1] For SIGN, we use a five-layer SIGN with a hidden size of 512 followed by a two-layer MLPs as classifier and another two-layer MLPs as link predictor with hidden features of 512. [2]

**Results.** Our experiments are conducted over ten runs with different random seeds, and the final results are reported as the average accuracy in Table 5. We observe a constant performance improvement of our method compared to the baseline. Specifically, our approach achieves 1.74% and 0.62% accuracy improvement on GraphSAGE and SIGN, respectively.

## 5 CONCLUSIONS

In this work, we introduce GRAPHDET, a framework designed to enhance graph domain adaptation performance. It uses auxiliary edge tasks, such as denoising target graphs, to achieve this objective. Our experiments demonstrate superior performance compared to existing baselines in both time and regional graph domain adaptation. Additionally, we provide a theoretical justification, offering insights into the role of auxiliary edge tasks in tightening the graph domain adaptation bound.

For future directions, given these insights, it would be interesting to explore the extension of auxiliary edge tasks in graph domain adaptation. In real-world applications, particularly in online learning scenarios where graphs are continuously collected over time and instant labels are difficult to obtain, applying domain adaptation algorithms in an online learning setting presents a critical and emerging challenge.

---

[1]Following: https://github.com/dmlc/dgl/tree/master/examples/pytorch/ogb/ogbn-products/graphsage
[2]Following: https://github.com/dmlc/dgl/tree/master/examples/pytorch/ogb/sign

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

# A  PROOF FOR PROPOSITION 3.1

First, for edge tasks, we consider that their objective is to compute heuristic node similarity scores, representing the likelihood of links (Liben-Nowell and Kleinberg, 2003; Kumar et al., 2020). GNNs can be used to learn the node embeddings, which are used to perform link prediction tasks (Zhang and Chen, 2018). With a slight abuse of the objective targets, we can consider one of the goals of these edge tasks to be ensuring that the embeddings of connected nodes have high similarity, denoted as:

$$\|x_u - x_v\| \le \xi_1, \tag{10}$$

where $\mathcal{E}_T$ represents the edges in the target graphs, and the node embeddings $x_u$ and $x_v$ denote the embeddings of the nodes at the endpoints of the edge after being processed by the GNN model $f(\mathcal{G}_T)$. In this case, the link tasks aim to tighten $\xi_1$ and encourage connected nodes to learn closer node embeddings. Meanwhile, the node embeddings will also be used for the subsequent classification task with a classifier $g$. We assume that the classifier is lipschitz continuous with a constant $L$, a commonly adopted assumption (You et al., 2023). Therefore, we have:

$$\|g(x_u) - g(x_v)\| \le L \|x_u - x_v\| \le L\xi_1. \tag{11}$$

Next, we first recall the generalization bound in Eqn. equation 4 and the definition of these two classifiers $g_1$ and $g_2$. We then apply this on the embeddings of the connected nodes $x_u$ and $x_v$. Instead of directly analyze, we consider the disagreement score value $d(g_1(x), g_2(x)) = \|g_1(x) - g_2(x)\|$ measuring the difference on the output of $g_1(x)$ and $g_2(x)$. As the classifiers $g_1$ and $g_2$ are $L_1$ and $L_2$-lipschitz continuous, we have:

$$
\begin{aligned}
&|d\left(g_1(x_u), g_2(x_u)\right) - d\left(g_1(x_v), g_2(x_v)\right)| \\
&\le |\, \|g_1(x_u) - g_2(x_u)\| - \|g_1(x_v) - g_2(x_v)\| \,| \\
&\le \|(g_1(x_u) - g_2(x_u)) - (g_1(x_v) - g_2(x_v))\| \\
&\le \|(g_1(x_u) - g_1(x_v)) + (g_2(x_u)) - g_2(x_v))\| \\
&\le L_1\|x_u - x_v\| + L_2\|x_u - x_v\| \\
&\le (L_1 + L_2)\xi_1,
\end{aligned}
\tag{12}
$$

with the reverse triangle inequality. Then, we perform this on all the nodes instead of edges, we have:

$$
\begin{aligned}
&d\left(g_1(x_u), g_2(x_u)\right) \\
&\le \min\left\{ \min_{v \in \mathcal{N}_u} d\left(g_1(x_v), g_2(x_v)\right) + (L_1 + L_2)\xi_1, d\left(g_1(x_u), g_2(x_u)\right) \right\} \\
&\le d\left(g_1(x_{u^*}), g_2(x_{u^*})\right) + K(u, u^*)(L_1 + L_2)\xi_1,
\end{aligned}
\tag{13}
$$

where $u^* = \arg\min_{u' \in \mathcal{C}_u(\mathcal{G}_T)} d\left(g_1(x_{u'}), g_2(x_{u'})\right) s.t. u \in \mathcal{C}_u(\mathcal{G}_T)$ denotes the node that have the lowest disagreement score in the graph component $\mathcal{C}_u(\mathcal{G}_T)$ which contains node $u$. $\mathcal{N}_u$ denotes the neighbors of node $u$. Additionally, $K(u, u^*)$ represents the hop distance between node $u$ and $u^*$.

When it comes to $\Pr_{D_T}[\mathcal{Z}_{g_1}\Delta\mathcal{Z}_{g_2}]$, which is the upper bound in Eqn. equation 4, it can be approximated by the disagreement scores of all nodes within the target graphs. We assume that when the disagreement scores are smaller than a constrain $\xi_2$, the $g_1$ and $g_2$ tend to agree on the final prediction with probability $1 - \delta$ on $D_T$. Formally, it can be expressed as follows:

$$\Pr_{D_T}\left(Z_{g_1}\Delta Z_{g_2}\right) \le \frac{1}{\xi_2}\mathbb{E}_{x_u \sim D_T}\left[d\left(g_1\left(x_u\right), g_2\left(x_u\right)\right)\right], \tag{14}$$

with $1 - \delta$ probability. When we perform link tasks to constrain the $\xi_1$, the overall disagreement scores on target will also be constrained, forcing the domain adaption bound to be tighter.

Summary is shown here:

- Our work focuses on analyzing and constraining the generalization bound in this context. We do not explicitly match the nodes on source or target graphs, but constrain the GNN to obtain the node embeddings in which a classifier, denoted as $g_1$, trained with source graph embeddings and source graph labels can have comparable performance compared to

the classifier, denoted as $g_2$, trained using target graph embeddings and target labels when testing. In particular, we only have source graph labels to train $g_1$, but we can make the performance of $g_1$ on target graphs become closer to $g_2$, that is, achieve better generalization ability.

- We first find the node on the target graph that has the most similar results under $g_1$ and $g_2$, denoted as $\mathbf{x}_{u^*}$. This is a bridge node on $g_1$ and $g_2$, and the probability of making a different prediction is shown as $d(g_1(\mathbf{x}_{u^*}), g_2(\mathbf{x}_{u^*}))$. Then, we need to extend it to all the nodes on target graphs, shown as $K_{\text{avg}}(C_{u^*})(L_1 + L_2)\xi_1$, by using the assumption that $g_1$ and $g_2$ are $L_1$- and $L_2$-Lipschitz continuous. Since we only need to extend it to all the nodes on target graphs, we just apply this loss function on target graphs. While the term $\xi_1$ becomes the similarity of node embeddings on the target edges, the proposed method can help improve the generalization ability of graph domain adaptation by constraining $\xi_1$.

# B  ABLATION STUDIES

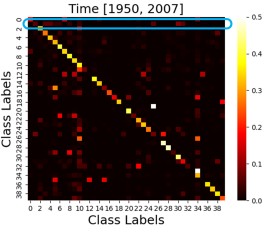
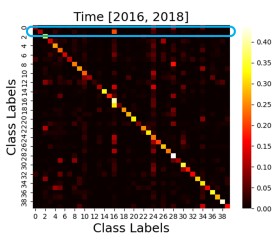
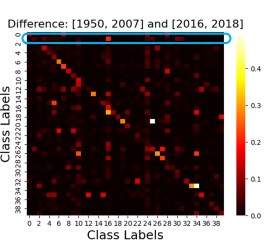

(a) From 1950 to 2007.

(b) From 2016 to 2018.

(c) The difference in label connectivity relationships between the previous two graphs.

Figure 2: Figures show the normalized label connectivity relationship for the graphs extracted on Arxiv with time period 1950-2007, 2016-2018, and the difference of between these two time periods, highlighting the structural changes over time.

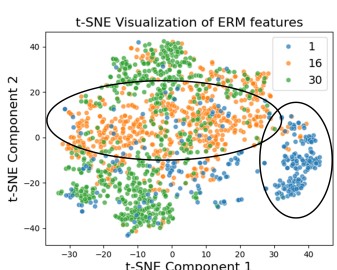
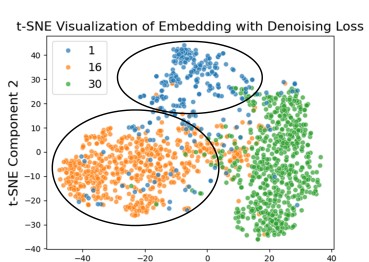

(a) Node embedding visualization for class 1 class 16 and class 30 on target graphs Arxiv (2016-2018). Note that model is trained via ERM on source graph Arxiv (1950–2007).

(b) Node embedding visualization for classes 1, 16 and 30, trained on the Arxiv (1950–2007) with an auxiliary denoising target graph task on target graphs Arxiv (2016–2018).

Figure 3: Comparison of visualization of node embeddings on target graphs Arxiv (2016–2018).

The visualizations of node embeddings on target graph from the previous example are shown in Figure 3. When training GNNs solely by minimizing the empirical risk, we observe that while nodes of class 1 (denoted in blue in Figure 3a) tend to form clusters highlighted by circle, many are scattered throughout the entire figure. Notably, the blue points are scattered among the orange and green points, indicating that the node embeddings of class 1 may be similar to those of other classes. As observed in the label connectivity relationship, class 1 should be disconnected from class 30. However, node embeddings from these two classes exhibit substantial overlap. In Figure 3b, we present the node embeddings on target graph using our framework. Here, most class 1 nodes are

Table 6: Performance comparison on graph domain adaptation tasks. Best results in each column are in **bold**, and underline denotes the second best performance.

| Models | Airport | | Blog | | ArnetMiner | |
|---|---|---|---|---|---|---|
| | E→U | U→E | B1→B2 | B2→B1 | C→A | D→A |
| DANE | $31.18 \pm 3.26$ | $33.75 \pm 0.31$ | $\underline{32.17 \pm 3.20}$ | $32.77 \pm 0.66$ | $62.87 \pm 1.40$ | $59.19 \pm 1.66$ |
| DGDA | $43.45 \pm 2.16$ | $43.78 \pm 2.90$ | $22.10 \pm 1.06$ | $21.06 \pm 2.20$ | $52.20 \pm 2.46$ | $56.31 \pm 2.04$ |
| PairAlign | $42.38 \pm 0.77$ | $36.84 \pm 1.48$ | $\underline{32.17 \pm 10.88}$ | $\underline{41.16 \pm 0.86}$ | $58.06 \pm 2.62$ | $56.68 \pm 0.89$ |
| SpecReg | $37.59 \pm 2.55$ | $28.91 \pm 8.77$ | $28.27 \pm 4.22$ | $30.30 \pm 1.35$ | $68.90 \pm 4.78$ | $66.30 \pm 4.28$ |
| A2GNN | $\mathbf{50.64 \pm 1.47}$ | $\underline{53.47 \pm 0.24}$ | $22.58 \pm 0.01$ | $33.04 \pm 4.12$ | $\mathbf{76.15 \pm 0.16}$ | $\mathbf{74.12 \pm 0.18}$ |
| GraphDeT | $\underline{50.14 \pm 0.39}$ | $\mathbf{55.47 \pm 0.66}$ | $\mathbf{52.20 \pm 0.51}$ | $\mathbf{46.19 \pm 8.54}$ | $\underline{73.91 \pm 0.22}$ | $\underline{70.85 \pm 0.15}$ |

well-clustered within the circle, with only a few overlapping with class 30. Some nodes are placed near class 16, which may be reasonable since classes 1 and 16 are densely connected in the target graphs shown in Figure 2b.

Here, we can have an intutive example. We take a three classes with node features 1, $\delta$ and 0. Previously, it is hard to distinguish $\delta$ and 0 in source graphs. But with the node embedding learned with link prediction on target graphs, which contains lots of connection between class 1 and class 2. The node belongs to class 2 can be distinguished with class 3. While in training graphs, it is hard to distinguish these two as these two are connected together without connection with class 1.

## C  EXPERIMENTAL RESULTS ON PYGDA.

We follow the same experimental setting as the pygda (Liu et al., 2024b) benchmark. And the propose GraphDeT achieves the best performance on three tasks, and second best performance on the other three tasks.

