# OpenReview forum: "Empowering GNNs for Domain Adaptation via Denoising Target Graph"
_ICLR.cc/2026/Conference — ICLR 2026 Conference Withdrawn Submission_

### Official Review · Reviewer_fAb9 · 2025-10-28

**Soundness:** 2
**Presentation:** 3
**Contribution:** 3
**Rating:** 4
**Confidence:** 3

**Summary:**

This paper addresses the problem of unsupervised graph domain adaptation for node classification, where a Graph Neural Network trained on a labeled source graph needs to generalize to an unlabeled target graph with a different data distribution. The authors propose a simple yet highly effective framework called GRAPHDET. The core idea is to introduce an auxiliary self-supervised task on the unlabeled target graph: Perform the main node classification task on the labeled source graph and an auxiliary edge denoising task on the target graph. This is done by adding random, "noisy" edges to the target graph and training an edge predictor to distinguish the original edges from the fake ones.

**Strengths:**

1. The proposed method is simple to implement—it essentially adds a link prediction-style loss on the target graph—yet it yields large and consistent performance improvements over a range of strong baseline.

2.  The experimental setup is comprehensive. The paper uses standard, challenging datasets for time-based (Arxiv) and regional (MAG) domain shifts. The comparison includes recent and relevant baselines like SPECREG and Pairwise Alignment (PA). The ablation study in Table 4 also effectively justifies the choice of the denoising loss (DeT) over other variants like GAE or standard link prediction.

**Weaknesses:**

1. The method introduces a new loss term ($l_{DeT}$), which must be balanced with the primary classification loss ($l_{cls}$), presumably with a weighting hyperparameter (e.g., $\lambda$). This is a critical detail for reproducibility and understanding the method's robustness. The paper does not mention this hyperparameter, how it was selected, or how sensitive the model's performance is to its value. A sensitivity analysis is a key missing piece.

2. The connection between the practical loss function (Eq. 7, binary cross-entropy for denoising) and the theoretical bound (Eq. 10, $||x_u - x_v|| \le \xi_1$) could be made more explicit. Minimizing the denoising loss encourages the model to discriminate between real and fake edges, which is related to, but not mathematically identical to, minimizing the distance between embeddings of connected nodes. A more direct explanation of how the loss objective minimizes $\xi_1$ would strengthen the theoretical argument.

**Questions:**

See weakness section.

---

### Official Review · Reviewer_W2JB · 2025-10-30

**Soundness:** 2
**Presentation:** 2
**Contribution:** 2
**Rating:** 2
**Confidence:** 5

**Summary:**

This paper presents GraphDeT, a framework for Graph Domain Adaptation (GDA) that introduces an auxiliary edge-denoising task on the target graph. The key idea is that by adding and removing random edges, then training the model to distinguish true from false connections, the learned node embeddings become more robust to structural noise. The authors claim that this auxiliary task tightens the domain adaptation generalization bound through a reduction in A-distance, theoretically supported by a proof extending the classic Ben-David et al. (2006) bound.

**Strengths:**

* **Empirical performance** GraphDeT yields large and consistent performance gains across both temporal (Arxiv) and regional (MAG) adaptation benchmarks. The ablation on various edge-related tasks (GAE, link prediction, denoising) provides solid empirical grounding.
* **Practical simplicity** The proposed approach is simple to implement and can be readily integrated with existing GNN architectures.
* **Connection to theory** The authors attempt to bridge empirical improvements and theoretical insight by connecting their auxiliary task to domain adaptation theory.

**Weaknesses:**

1. **Motivation for denoising remains insufficient.**
   The paper does not convincingly explain *why* the target graph requires denoising in the first place. While Section 3.1 assumes the existence of “noisy edges,” the source of such noise, its statistical characteristics, or its relation to domain shift are never justified. Without concrete motivation, the denoising task risks appearing as an *ad-hoc* regularizer rather than a principled GDA mechanism.

2. **Theoretical novelty is limited.**
   *Theorem 2.1* directly replicates the standard Ben-David et al. (2006) bound without substantive extension. The subsequent Proposition 3.1 merely rephrases the Lipschitz continuity assumption to derive an inequality similar in spirit to prior analyses (e.g., You et al., ICLR 2023; Liu et al., ICML 2024a; Fang et al, ICLR 2025). Hence, the “extension” to structural constraints does not constitute genuine theoretical innovation.

3. **Insufficient engagement with recent GDA literature.**
   The related-work section omits several key recent papers that have advanced the field. In particular, *Chen W., Ye G., Wang Y., et al. (AAAI 2025),* “Smoothness Really Matters,” provides a simple and theoretically grounded approach that should be discussed and compared. Other missing works include more recent spectral and topology-aware methods. Their absence weakens the contextualization and novelty claim.

4. **Limited use of attribute information.**
   The framework exclusively denoises *edges* on the target graph but does not exploit *attribute-level* discrepancies (node features), which are often a dominant factor in domain shift. Without leveraging attribute alignment or feature-space adaptation, the proposed method may address only a subset of the GDA problem.

5. **Theoretical–empirical gap.**
   While the theory centers on constraining A-distance via embedding similarity, the experiments provide no quantitative evidence that A-distance or disagreement truly decreases. Including such empirical verification (e.g., estimating $d_{H\Delta H}$ or feature discrepancy before and after denoising) would make the theoretical claim more convincing.

In addition, there is a problem with the citation format in lines 146-147 of the article.

**Questions:**

Please See Weaknesses

---

### Official Review · Reviewer_gwqL · 2025-11-01

**Soundness:** 2
**Presentation:** 2
**Contribution:** 2
**Rating:** 4
**Confidence:** 3

**Summary:**

This paper studies node classification problem under graph structure shift. Based on the finding that adding an edge task on the target graph substantially improves target performance, the authors propose GRAPHDET, which jointly optimizes the source node classification loss and a target-edge denoising loss. Theoretical analysis and experimental results demonstrate the effectiveness of the proposed method.

**Strengths:**

1.	The proposed method is simple and easy to implement yet yields notable performance improvement in both time and regional domain adaptation scenarios.
2.	The authors establish a connection between the auxiliary edge task and generalization bound and provide a proof sketch suggesting that enforcing embedding similarity across edges can, under assumptions though, reduce classifier disagreement and thus tighten the A-distance term in the classical domain adaptation bound.

**Weaknesses:**

1.	The proposed method leverages only label supervision from the source domain but does not transfer or model source structural patterns. Thus, the method does not explicitly learn how source structural priors can generalize to target graphs. This raises questions about whether the method truly performs domain adaptation, i.e., leveraging knowledge from the source domain, or simply applies target side self-supervised regularization.
2.	There remains some disconnection between the theoretical analysis and the proposed method. While the auxiliary task may improve structural representations and overall generalization, it is unclear whether the method genuinely reduces classifier disagreement or domain discrepancy as claimed. The observed gains might stem from general regularization effects of target-only self-supervision rather than from cross-domain alignment. If this is the case, the contribution as a domain adaptation technique is weakened.
3.	The effectiveness of the proposed method may be limited to scenarios with relatively small domain shifts. The experiments are conducted on datasets representing different time periods or geographical regions of the same domain. In such settings, structural differences between source and target graphs may be minor, and the label–structure relationships likely remain consistent. In cases of larger cross-domain gaps, the generalizability of the proposed method remains uncertain.

**Questions:**

1.	The two tasks are optimized jointly with equal weights. Could their objectives conflict during training? How would varying the weight between the classification and denoising losses affect performance?

2.	While the edge-denoising task can indeed help the model learn meaningful target graph structure, how does this relate to structural shift? Moreover, the A-distance in classical theory usually captures feature distribution discrepancy rather than structural discrepancy. How exactly does structure shift influence the A-distance or classifier disagreement in this context?

3.	The paper mentions that making all embeddings identical would trivially tighten the bound but eliminate useful information. How does the proposed method prevent such embedding collapse?

---

### Official Review · Reviewer_zF7X · 2025-11-03

**Soundness:** 2
**Presentation:** 2
**Contribution:** 2
**Rating:** 2
**Confidence:** 3

**Summary:**

The authors introduces GraphDeT, a framework designed to enhance the domain adaptation ability of GNN across different temporal and regional domains. By incorporating auxiliary edge tasks, the approach emphasizes capturing meaningful structural relationships between nodes to improve generalization. Extensive experiments demonstrate that GraphDeT substantially outperforms existing methods. The analysis shows that the auxiliary edge tasks help address structural shifts and distributional biases, making the model more robust to domain changes. Overall, the study highlights the significant role of edge-based tasks in strengthening graph domain adaptation, with promising implications for practical applications involving evolving and diverse graph data.

**Strengths:**

1.	The results are significant and demonstrate clear performance gains.
2.	The authors provide a theoretical foundation, showing that auxiliary edge tasks effectively tighten the domain adaptation bounds. This theoretical insight offers a deeper understanding of their role in mitigating domain shifts.

**Weaknesses:**

1.	The Introduction section lacks clarity and persuasive motivation, and the proposed method does not exhibit sufficient novelty.
2.	The chosen baselines are outdated, with only one from 2024 studies, additional recent baselines should be included for a fair comparison.
3.	The Related Work section on graph domain adaptation only covers developments up to 2024 and cites merely one paper, making it incomplete and lacking in reference value. A more comprehensive literature review is needed.
4.	No code or implementation is provided, which raises concerns about reproducibility.
5.	The Method section is poorly structured and confusing in its presentation.

**Questions:**

see weakness.

---

### Note · Authors · 2025-11-21

I have read and agree with the venue's withdrawal policy on behalf of myself and my co-authors.